# International development and implementation of a core measurement set for research and audit studies in implant-based breast reconstruction: a study protocol

Shelley Potter [1,2] Charlotte Davies,[1] Christopher Holcombe,[3] Eva Weiler-Mithoff,[4] Joanna Skillman,[5] Raghavan Vidya,[6] Yazan Masannat,[7] Walter Weber,[8] Joerg Heil,[9] Sherif Wilson,[2] Steven Thrush,[10] Lisa Whisker,[11] Jane Blazeby,[1] Chris Metcalfe,[1] Kerry Avery [1]

For numbered affiliations see end of article.

**Correspondence to**
Shelley Potter;
shelley.potter@bristol.ac.uk

## ABSTRACT

**Introduction** Outcome reporting in research studies of breast reconstruction is inconsistent and lacks standardisation. The results of individual studies therefore cannot be meaningfully compared or combined limiting their value. A core outcome set (COS) has been developed to address these issues and identified 11 key outcomes to be measured and reported in all future research and audit studies in reconstructive breast surgery (RBS). A COS represents what key outcomes should be measured. The next step is to determine how and when this should be done. The aim of this study is to develop a core measurement set (CMS) for use in research and audit studies in implant-based breast reconstruction.

**Methods and analysis** The CMS will be developed in accordance with the guidance developed by the Core Outcome Measures in Effectiveness Trials initiative (COMET) and COnsensus-based Standards for the selection of health Measurement Instruments (COSMIN) group for the selection of outcome measurement instruments (OMIs) for relevant outcome domains included in the RBS COS. This will involve three phases with strategies to promote implementation as a final additional phase. The phases are (1) conceptual considerations in which the target population, procedures and settings are defined; (2) systematic reviews to identify existing clinical, patient-reported and cosmetic OMIs and, if appropriate, assess their quality using COSMIN methodology; (3) a modified Delphi process including sequential Delphi surveys involving approximately 100 healthcare professionals and a face to face consensus meeting to agree and ratify which outcome definitions and OMIs should be used and standardised time points for assessment; (4) strategies to promote dissemination and adoption of the CMS.

**Ethics and dissemination** Ethical approval has been granted by University of Bristol Faculty Research Ethics Committee FREC ID 60221. Dissemination strategies will include scientific meeting presentations and peer-reviewed journal publications. Implementation activities will include engagement with journal editors and funders to promote uptake and use of the CMS.

### Strengths and limitations of this study

► Robust consensus methods with international participation will agree standardised measures for use in all future research and audit studies of implant-based breast reconstruction which will improve the quality and comparability of future results.

► Candidate measures for inclusion in the Delphi process will be selected based on high-quality systematic reviews.

► This study will focus on generating consensus among healthcare professionals; robust patient and participant involvement will be needed to ensure that the patient-reported outcomes recommended in this study will be acceptable to patients undergoing implant-based reconstruction.

► Further work will be needed to determine the degree to which the planned implementation of the core measurement set has been successful.

## INTRODUCTION

Appropriate outcome selection is vital if research is to inform practice and guide policy. The development and use of core outcome sets (COSs), a scientifically agreed minimum set of outcomes that should be measured and reported in effectiveness studies in a given area is one way in which inconsistent and heterogeneous outcome selection may be addressed.[1]

A COS for reconstructive breast surgery (RBS) has recently been developed.[2] This study used robust Delphi consensus methodology with over 300 patients and healthcare professionals followed by consensus meetings to agree and ratify an 11-item COS.[2] The COS includes clinical (implant and flap-based complications, major complications and unplanned surgery),

patient-reported (quality of life, normality, emotional and physical well-being and self-esteem) and cosmetic (women's cosmetic satisfaction) outcomes that all stakeholders felt were important to measure as a minimum in future effectiveness studies in RBS.

While a COS is an important step in determining *what* outcomes should be measured in research and audit studies in RBS, it does not describe *how* these key outcomes should be measured. The next step in improving the quality and consistency of outcome reporting in RBS is therefore to develop a core measurement set (CMS), a set of instruments to assess the COS and to generate consensus regarding the standard time points at which these outcomes should be measured.[3–5] The need for a CMS to improve the quality and value of research in oncoplastic and RBS was identified as a research priority in the recently published breast surgical gap analysis.[6]

The COS was developed for RBS as a whole but this includes a diverse range of procedures ranging from total breast reconstruction following mastectomy with implants or tissue-based procedures and oncoplastic breast conservation using both volume replacement and displacement techniques. The COS therefore includes procedure-specific domains such as 'implant-related complications' and 'donor-site morbidity' that would not be applicable to all types of reconstruction. Furthermore, the types of complications that are relevant for specific procedure types will differ. For a CMS to be meaningful and easily applicable in practice, a separate set of core measures for each procedure group will be needed. As implant-based breast reconstruction (IBBR) is the most commonly performed procedure worldwide[7 8] and a rapidly evolving area of clinical practice with new procedures and techniques such as prepectoral reconstruction currently being introduced and evaluated,[9] development of a CMS for IBBR is a research priority. It is anticipated that CMS for autologous breast reconstruction and oncoplastic breast conservation will be developed in the future using similar methods.

Outcome assessment and reporting in IBBR will only be improved if the resultant CMS is accepted and adopted into practice.[5] Studies to date have reported variable success in the implementation of COS in other areas.[10–12] Reasons for this are complex but optimising 'buy in', for example, by involving appropriate and geographically diverse stakeholder groups in the development process and using a solid dissemination strategy have been identified as important.[10] Effective strategies to promote dissemination and implementation will therefore be necessary to increase awareness of the CMS in the reconstructive community and promote its uptake and use in future research studies.[5]

## Aim
The aim of the study is to develop a CMS for IBBR and develop strategies by which awareness of the CMS and its subsequent adoption and implementation may be enhanced.

## METHODS AND ANALYSIS
### Overview
The development of the CMS will be based on the guidance developed by Core Outcome Measures in Effectiveness Trials (COMET) and COnsensus-based Standards for the selection of health Measurement Instruments (COSMIN) on the selection of outcome measurement instruments (OMIs) for the outcome domains included in the RBS COS.[13] This involves three stages with strategies to promote implementation as a final additional phase.
1. Conceptual considerations and scope of the CMS.
2. Systematic reviews to identify existing OMIs and assess their quality.
3. An international Delphi process to establish consensus among stakeholders regarding which complications, definitions and OMIs should be used and standardised time points for assessment and consensus meetings to agree and ratify these decisions.
4. Strategies to promote dissemination and adoption of the CMS.

### Patient and public involvement
Patients and members of the public will be involved throughout the study. A patient and public involvement (PPI) group will be established as part of the project and a patient will be invited to sit on the study steering group. They will contribute to all study phases and dissemination of the results.

### Phase 1. Conceptual considerations and scope of the CMS
The first step in CMS development is to agree the construct (ie, outcome or domain) to be measured and the target population.

The constructs were identified in the original COS study[2] and include 11 domains, 9 of which are relevant to IBBR. These comprise three clinical (major complications, unplanned surgery for any reason and implant-related complications) and six patient-reported (normality, quality of life, women's cosmetic satisfaction, physical well-being, emotional well-being and self-esteem) outcome domains. The two remaining core outcome domains, flap-related complications and donor-site morbidity are not relevant to IBBR so will not be included in this process.

The target population for the CMS will be adult women undergoing immediate or delayed breast reconstruction surgery using implants or expanders placed in any position (subcutaneous or submuscular) with or without mesh following mastectomy for breast cancer or risk reduction.

The COS was developed for use in research and audit studies of RBS and the CMS will similarly be developed for use in any clinical evaluation of IBBR to optimise the value of this work and allow the results of all future studies to be compared and combined.

An expert steering group comprising breast and plastic surgeons, clinical nurse specialists and patients will be convened to oversee the study, review the results of the

systematic review, design the Delphi survey and act as champions for the adoption and dissemination of the CMS in the breast reconstruction community.

## Phase 2. Identification of existing OMIs and OMI quality assessment using systematic reviews

The term 'OMI' will be used to refer to the method by which the outcome is being measured (the tool used to assess the outcome) and may be a single question, a questionnaire or other appropriate tool depending on the outcome to be assessed.[3]

Systematic reviews will be used to identify existing measurement instruments for each of the relevant IBBR core outcomes and to identify gaps where new instruments may be needed. The review has been registered on the PROSPERO International prospective register of systematic reviews (CRD42017075211).

### Clinical outcomes

The existing systematic review of the clinical outcomes of breast reconstruction[14] will be updated to identify all additional published randomised clinical trials (RCTs) or RCT protocols and non-randomised comparative or non-comparative studies or published protocols reporting outcomes of or planning to recruit a minimum of 100 patients undergoing IBBR following mastectomy using any technique. A minimum sample size of 100 patients was used in the original systematic review to restrict focus to studies that would be sufficiently large to influence practice. Ongoing clinical trials will be identified from clinical-trials.gov and primary and secondary outcome measures, outcome definitions and time point of outcome assessments extracted.

Abstracts will be screened by one reviewer (SP) to identify eligible studies and outcome definitions and time point of outcome assessments will be extracted verbatim with checking of a proportion by a second reviewer (CD). The individual outcomes and definitions will be grouped according to core clinical outcome domain (implant-based complications, major complications, unplanned surgery for any reason) for use in phase 3, the Delphi survey.

Outcomes, definitions, time points for outcome assessments and composition of outcome domains will be reviewed by the expert steering group for completeness and relevance prior to progression to the Delphi process.

### Patient-reported outcomes

A structured search of MEDLINE, EMBASE and PsycINFO will identify systematic reviews of OMIs for relevant patient-reported outcome domains (health-related quality of life (HRQL), emotional well-being,; physical well-being, normality, self-esteem, cosmetic satisfaction). If systematic reviews are of high quality and have undertaken a quality assessment of the OMIs, then one OMI will be selected and presented to a group of key stakeholder representatives at the end of the process. If there are no systematic reviews or they are of poor quality, a new or updated systematic review of the measurement properties

of patient-reported outcome measures (PROMs) developed for or validated in women undergoing RBS will be performed.[15 16]

A systematic search will be performed in MEDLINE and EMBASE using search terms relating to the construct of interest (PROMs) and the target population (IBBR) combined with the measurement properties filter described by Terwee *et al.*[17] Primary studies reporting the development or validation of PROMs in women undergoing RBS will be eligible for inclusion with particular reference to studies evaluating instruments relevant to each of the core patient-reported outcomes (HRQL, normality, women's cosmetic satisfaction, physical well-being, emotional well-being or self-esteem). Only full-text papers published in English will be included in the review. Eligibility assessments and data extraction will be performed by two reviewers (SP, CD) and discrepancies resolved by discussion with a third reviewer (KA). Tables will be constructed to summarise included study characteristics, instrument characteristics, measurement properties and interpretability. The methodological quality of included studies will be assessed using the COSMIN checklist.[18] Best evidence synthesis will be undertaken if more than one study has assessed a particular measurement property. For each instrument identified in the review, recommendations will be made regarding its potential applicability to the CMS or the need for further validation work. The most appropriate instrument for each core outcome will be identified. If more than one instrument is considered equally valid, details of each potential instrument will be included in the Delphi process to determine which tool should preferentially be used in the IBBR–CMS. If no suitable instrument is identified for any of the core outcomes, work will be undertaken with key stakeholders to develop and validate a new tool according to the guidelines reported in the COSMIN checklist.[16]

### Cosmetic outcomes

The cosmetic outcomes identified in the COS are patient reported, so identification of appropriate measures will be addressed in the PROM systematic review.

## Phase 3. Delphi process to establish consensus regarding OMI section, definitions and time point

The Delphi process will consist of initial questionnaire development based on the results of the systematic reviews, sequential questionnaire administration and a final consensus meeting with the aim of generating stakeholder agreement regarding the final CMS.

### Questionnaire development

The complications, definitions and time points identified from the updated clinical systematic review and details relating to appropriate PROMs for example, number and content of items for each core outcome will be reviewed. Only complications reported in more than 10% of studies in the systematic review will be included in the Delphi together with the most commonly reported definitions and

outcome assessment time points. These will be operationalised and formatted into items for the Delphi survey which will be used to generate consensus for the IBBR CMS.[19 20] It is anticipated that the survey will have four sections to determine: (1) what complications should be included in the 'implant-related complications' domain; (2) the most appropriate standardised definitions of complications to be used in the final CMS; (3) the most appropriate validated PROM (if any) for measuring each of the patient-reported outcome domains and (4) the most appropriate time point for short-term and long-term clinical and patient-reported outcome assessment in future clinical studies.

The items for section (1) to be included in the implant-related complication domain, will be rated on a 9-point Likert scale from 1 (not important) to 9 (essential) based on the Grading of Recommendations Assessment, Development and Evaluation scale for scoring the importance of including the item in a COS as per established methodology.[21] For section (2) selection of the most appropriate outcome definitions and section (4) selection of the most appropriate time points for outcome assessment, the most commonly reported definitions and time points identified from the systematic reviews will be presented and respondents asked to select the most appropriate definition or time point for each outcome. Finally for section (3) details of any validated PROM identified for each patient-reported outcome (PRO) domain will be presented and respondents asked if (1) the PROM is appropriate and (2) if more than one validated PROM is available, to rank the instruments from best (most appropriate) to worst (least appropriate).

The structure and content of the Delphi survey will be reviewed and discussed by the members of the steering group to ensure that the candidate complications, definitions and time points are complete, relevant and practical prior to the piloting phase.

The questionnaire will be piloted with a group of breast and plastic surgeons to test understandability and acceptability prior to its use in Delphi round 1.

### Selection and recruitment of study participants

Key stakeholders including surgeons performing IBBR and specialist nurses involved in counselling patients undergoing IBBR will be invited to complete the Delphi survey.

Patients will not be involved in the Delphi process as they do not have the technical knowledge to select the most appropriate complications or definitions for use in the CMS. Patients, however, will be involved in the steering group and results of the Delphi process, in particular the PROMs and proposed time points for outcome assessment, will be discussed with a PPI focus group to ensure that they are acceptable and meaningful to patients.

Multi-national professional stakeholders will be invited to participate via the professional associations including the UK Association of Breast Surgery and British Association of Plastic Reconstructive and Aesthetic Surgeons; and international collaborative groups including the Oncoplastic Breast Consortium, personal research networks and social media. Invitations will include a link to an electronic survey.[22] The first page of the survey will include an invitation letter explaining the background and purpose of the study. The Delphi process will be explained and the importance of participating in each round emphasised.

### Delphi survey rounds

Participants will complete up to three sequential rounds of the Delphi survey over a 3-month period. In each round, participants will be asked to score the importance of including each identified complication in the implant-related complications domain and select the most appropriate outcome definitions, PROMs and time points for outcome assessment. All surveys will be administered online using the secure REDCap electronic data capture software.[22]

Participants who complete round 1 will be sent the round 2 survey. The second survey will contain all items retained from round 1 (see Data analysis section) with anonymised feedback from the previous round in the form of summary scores (eg, median scores for complications, proportions of participants selecting each definition/PROM/time point). Graphical representations will be used if appropriate to aid data visualisation. Participants will be asked to rescore each item based on the feedback received. If there is insufficient consensus after round 2, a further round may be conducted. This will be methodologically identical to round 2. Items retained after the final Delphi round will be carried forward to the consensus meeting.

### Consensus meetings

A purposive sample of professionals who participated in the Delphi survey will be invited to attend a face to face consensus meeting to discuss and agree the final CMS.

A summary of the survey results will be presented. Participants will be asked to ratify inclusion/exclusion of complications/measures during the Delphi survey and anonymously revote on complications/measures for which consensus was not reached during the Delphi or for which there were disagreements (see Data analysis section). Moderated discussion and revoting will be undertaken as necessary until consensus is achieved.

PPI focus groups will be held with patients to discuss OMI selection based on content, response burden and time point of PROMs assessment.

### Sample size

There is no standardised methodology for sample size calculation in CMS development but as with COSs,[23] the aim is to obtain good representation from key stakeholder groups. IBBR in the UK is predominantly performed by breast surgeons with plastic surgeons performing fewer procedures. Internationally, RBS is also performed by surgical oncologists and gynaecologists. A pragmatic approach[24] will therefore be taken to ensure adequate representation of surgeons performing the technique. It is anticipated that approximately 100 professionals will be recruited to the study. The majority of the Delphi items refer to technical surgical issues so it is not considered appropriate to

**Table 1** Definitions of consensus and management of items between rounds

| Category | Definition | Action |
|---|---|---|
| Items for inclusion in 'implant-related complication' domain | | |
| Consensus in | Scored as very important (7–9) by ≥70% and not important (1–3) by <15% of respondents | Item retained for next survey round/consensus meeting |
| Consensus out | Scored as not important (1–3) by ≥70% and very important (7–9) by <15% of respondents | Item discarded after round 2 (to be ratified at consensus meeting). |
| No consensus | Criteria above not met | Item retained for next survey round/consensus meeting |
| Selection of outcome definitions | | |
| Consensus in | Definition selected by ≥75% respondents | Definition retained for next survey round/consensus meeting |
| Consensus out | Definition selected by <5% respondents | Definition dropped after round 2 |
| No consensus | Criteria above not met | Definition retained for next survey round/consensus meeting |
| Selection of patient-reported outcome measures | | |
| Consensus in | Scored as 'best' by ≥75% and 'worst' by <15% of respondents | PROM retained for next survey round/consensus meeting |
| No consensus | Criteria above not met | PROM retained for next survey round/consensus meeting |
| Selection of time point of outcome assessment | | |
| Consensus in | Time point selected by ≥75% respondents | Time point retained for next survey round/consensus meeting |
| Consensus out | Time point selected by <5% respondents | Time point dropped after round 2 |
| No consensus | Criteria above not met | Time point retained for next survey round/consensus meeting |

involve patients in this process. A separate PPI group will be convened at which candidate PROMs and proposed time point of assessment will be presented and discussed to ensure these are appropriate and important to patients.

### Data analysis
*Retaining or dropping items between survey rounds*
Data will be collected using REDCap,[22] a mature, secure web application for building and managing online surveys and databases hosted at Bristol Medical School https://sscmredcap.bris.ac.uk/redcap/ and analysed using STATA 14 MP (www.stata.com).

Descriptive statistics will be used to summarise the result from round 1 including the summary scores for each complication (eg, median, range) and the proportion of participants rating each complication as very important (score 7–9), equivocal (scores 6–4) or not important (scores 1–3). The percentage of respondents selecting each candidate definition and time point of outcome assessment will be calculated. For ranked items (PROMs), the percentage of respondents rating each item best and worst will be recorded. All items will be retained between rounds 1 and 2. After round 2, items will be categorised as described in table 1. A third round will be held if the number of items for which no consensus is achieved following round 2 is too large for discussion at the consensus meeting (eg. >30). Round 3 will be analysed using the same criteria as described in table 1.

Complications/definitions/PROMs/time points for which there is disagreement by predetermined criteria (table 1) will be carried forward for discussion and voting at the consensus meeting.

*Consensus meeting*
Following the first round of voting, items will be categorised as 'consensus in', 'consensus out' or 'no consensus' as per table 1. Items voted 'consensus in' will be included in the CMS; items voted 'consensus out' will be discarded and items for which there is 'no consensus' will be rediscussed. Further rounds of voting and discussion will take place until consensus is achieved. The consensus meeting will conclude with participants ratifying the final CMS.

### Phase 4. Strategies to promote awareness, uptake and adoption of the CMS
The implementation phase will use a range of methods to raise awareness of the CMS and promote its adoption as recommended by the COMET initiative group.[1] This will include publication of the CMS and consensus statement in a leading journal; presentation at breast and plastic surgical meetings; dissemination to journal editors, professional associations, clinical guideline developers, Cochrane reviewers and funders requesting that all future studies, reviews, publications or applications in IBBR include the CMS.[1] Guidance material will be prepared and made freely available to aid researchers in adoption of the CMS. Use of the CMS will be monitored at reviewed 5 years following the publication to evaluate the degree to which these implementation strategies have been successful.[5]

### ETHICS AND DISSEMINATION
Dissemination strategies will include scientific meeting presentations, peer-reviewed journal publications and plain English summaries for patient and public dissemination. Social media will also be used to raise awareness of

the CMS in the wider international reconstructive community. Implementation activities will include engagement with journal editors and funders to promote uptake and use of the CMS and patient groups and charities to raise awareness of the CMS and its importance.

**Author affiliations**
[1]Bristol Centre for Surgical Research, Population Health Sciences, Bristol Medical School, Canynge Hall, Bristol, UK
[2]Bristol Breast Care Centre, Southmead Hospital, Bristol, UK
[3]Linda McCartney Centre, Royal Liverpool and Broadgreen University Hospital, Liverpool, UK
[4]Canniesburn Department of Plastic Surgery, Glasgow Royal Infirmary, Glasgow, UK
[5]Department of Plastic Surgery, University Hospitals Coventry and Warwickshire NHS Trust, Coventry, UK
[6]New Cross Hospital, Royal Wolverhampton Hospitals NHS Trust, Wolverhampton, UK
[7]Breast Unit, Aberdeen Royal Infirmary, Aberdeen, UK
[8]University Hospital and University of Basel, Basel, Switzerland
[9]University Hospital Heidelberg, Heidelberg, Germany
[10]Breast Unit, Worcester Royal Hospital, Worcester, UK
[11]Nottingham University Hospitals NHS Trust, Nottingham, UK

**Contributors** SP conceived the study, wrote the protocol and obtained funding. SP, KA, JB, CD, CH, EWM, JS, SW, LW and ST contributed to the study design. CM provided statistical support and KA provided outcomes and methodological expertise. SP wrote the first draft of the manuscript. WW, YM, RV and JH facilitated international collaboration. All authors reviewed and critically revised the protocol and manuscript prior to submission.

**Funding** This study is funded by the NIHR as part of a Clinician Scientist Award (CS-2016-16-019) and supported by the NIHR Biomedical Research Centre at University Hospitals Bristol NHS Foundation Trust and the University of Bristol.

**Disclaimer** The views expressed are those of the authors and not necessarily those of the UK National Health Service, National Institute for Health Research or Department of Health.

**Competing interests** None declared.

**Patient consent for publication** Not required.

**Ethics approval** Ethical approval for the study has been granted by University of Bristol Faculty Research Ethics Committee FREC ID 60221. Completing the Delphi survey will be taken as implied consent from participating professionals, but written consent will be obtained prior to the consensus meeting.

**Provenance and peer review** Not commissioned; externally peer reviewed.

**ORCID iDs**
Shelley Potter http://orcid.org/0000-0002-6977-312X
Kerry Avery http://orcid.org/0000-0001-5477-2418

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
