## [Reviewer comments · BMJ Open]

ARTICLE DETAILS

TITLE (PROVISIONAL)	International development and implementation of a core measurement set for research and audit studies in implant-based breast reconstruction: A study protocol
AUTHORS	Potter, Shelley; Davies, Charlotte; Holcombe, Christopher; Weiler-Mithoff, Eva; Skillman, Joanna; Vidya, Raghavan; Masannat, Yazan; Weber, Walter; Heil, Joerg; Wilson, Sherif; Thrush, Steven; Whisker, Lisa; Blazeby, Jane; Metcalfe, Chris; Avery, Kerry

VERSION 1 – REVIEW

REVIEWER	Mauro Barone Campus Bio-Medico university of rome
REVIEW RETURNED	21-Nov-2019

GENERAL COMMENTS	congratulation. it is a very interesting topic and well written manuscript
--

REVIEWER	Elisabeth Elder Westmead Breast Cancer Institute, Sydney University, Australia
REVIEW RETURNED	19-Dec-2019

GENERAL COMMENTS	This initiative is very important to be able to adequately compare surgical and patient reported outcomes after breast reconstruction. Having clear and consistent, globally agreed definitions for complications and other outcomes will streamline evaluation of new techniques and development of guidelines. The inclusion of validated instruments for patient reported outcomes is crucial for the success of the project. The methodology is sound, including systematic review and the well-established Delphi consensus process. The broad representation of interested parties including breast and plastic surgeons as well as specialist nurses and consumer representatives from various geographical areas are also key components to ensure successful implementations of the findings from the project.
---